# SemPLeS: Semantic Prompt Learning for Weakly-Supervised Semantic Segmentation

## Abstract

*Weakly-Supervised Semantic Segmentation (WSSS)* aims to train segmentation models using training image data with only image-level supervision. Since precise pixel-level annotations are *not* accessible, existing methods typically focus on producing pseudo masks for training segmentation models by refining CAM-like heatmaps. However, the produced heatmaps may only capture discriminative image regions of target object categories or the associated co-occurring backgrounds. To address the issues, we propose a *Semantic Prompt Learning for WSSS (SemPLeS)* framework, which learns to effectively prompt the CLIP space to enhance the semantic alignment between the segmented regions and the target object categories. More specifically, we propose *Contrastive Prompt Learning* and *Class-associated Semantic Refinement* to learn the prompts that adequately describe and suppress the image backgrounds associated with each target object category. In this way, our proposed framework is able to perform better semantic matching between object regions and the associated text labels, resulting in desired pseudo masks for training the segmentation model. The proposed SemPLeS framework achieves SOTA performance on the standard WSSS benchmarks, PASCAL VOC and MS COCO, and demonstrated interpretability with the semantic visualization of our learned prompts. The source codes are provided in the supplementary.

## 1 Introduction

Semantic segmentation aims to classify every pixel in images to identify object categories and the associated regions, which can benefit various applications in the real world (Ronneberger et al. (2015); Meyer & Drummond (2017); Zendel et al. (2022)). While promising results have been presented by fully-supervised approaches (Liang-Chieh et al. (2015); Long et al. (2015); Chen et al. (2017a;b); Zhao et al. (2017); Chen et al. (2018); Zhao et al. (2018)), collecting pixel-level annotations could be time-consuming and expensive, and therefore limits the scalability and practicality of fully-supervised methods. To address this issue, *Weakly-Supervised Semantic Segmentation (WSSS)* has emerged as an alternative approach to train segmentation models with only coarse or incomplete annotations such as bounding boxes (Khoreva et al. (2017)), scribbles (Lin et al. (2016)), points (Bearman et al. (2016)), or image-level labels. Among these annotation forms, *image-level labels* which indicate the presence or absence of certain object categories are commonly used due to the efficiency in data collection and the availability in various benchmark image datasets. Since precise annotations of object positions are *not* observed, learning to localize and segment object categories from image-level supervision is particularly challenging. Most existing methods (Chang et al. (2020); Wang et al. (2020); Ru et al. (2022); Xu et al. (2022a)) focus on producing pseudo ground truth masks by refining CAM-like heatmaps (Zhou et al. (2016); Selvaraju et al. (2017)) with class labels as discriminative supervision. Despite the shown efficacy, the produced pseudo masks may still miss relevant regions of target object categories and fail to cover the entire object. Furthermore, co-occurring backgrounds associated with certain object categories may also be falsely activated (*e.g.*, roads, or trees in an image of a car). Consequently, learning precise image regions that align with the semantics of target objects from weak supervision remains a challenging task.

With the rapid growth in the amount of visual and linguistic data in recent years, several vision-language models (Lu et al. (2019); Chen et al. (2020); Radford et al. (2021)) have been proposed to bridge the underlying semantics between images and text descriptions. Given that both the images and the associated text labels (category names) are available in the setting of WSSS, the un-

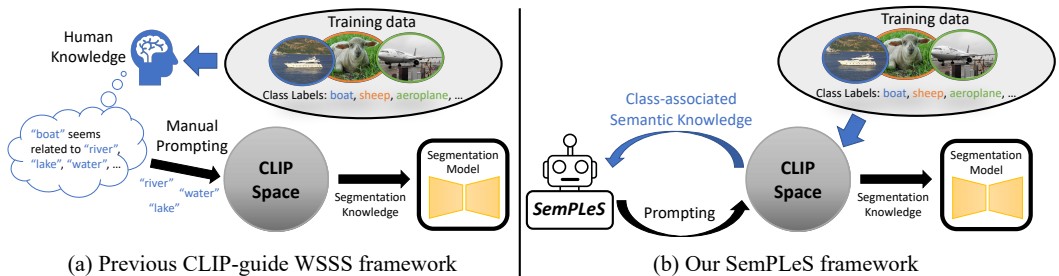

Figure 1: An overview of this paper. (a) Previous CLIP-guide WSSS methods require heuristic human knowledge involved with the additional manual prompt engineering effort, while (b) our proposed SemPLeS framework automatically learns class-associated prompts embedded with semantic knowledge discovered from the CLIP representation space without any manual human effort.

derlying image-text semantics from the CLIP (Radford et al. (2021)) space can be leveraged for pixel-level predictions. Recent approaches focus on manually designing text prompts to refine the produced pseudo masks, including background texts for suppressing co-occurring background regions (*e.g.*, {"a photo of river", "a photo of lake", "a photo of water"} for the "boat" category (Xie et al. (2022a)). Despite the effectiveness demonstrated, these approaches require heuristic human knowledge involved with the additional manual effort, as shown in Figure 1 (a). Moreover, manual prompting may not fully exploit the semantic knowledge inside the CLIP space.

In this paper, we aim to fully exploit the CLIP space to extract semantic information that can benefit the weakly-supervised semantic segmentation problem without manual prompting. To achieve this goal, we propose a ***Sem**antic **P**rompt **Le**arning for WSSS (**SemPLeS**)* framework to learn prompts embedded with the semantic knowledge *discovered* from the CLIP space, as shown in Figure 1 (b), where the learned prompts can enhance the semantic alignment between the segmented regions and the target object categories with image-level labels. More specifically, we perform image-text contrastive learning under the guidance of CLIP and train a mask generator to generate class activation maps. However, such produced masks might not be sufficiently precise, and the co-occurring backgrounds associated with the object categories may be falsely activated. To alleviate this problem, we uniquely present *Contrastive Prompt Learning* and *Class-associated Semantic Refinement* to suppress class-associated background regions. In *Contrastive Prompt Learning*, we learn prompts to capture co-occurring backgrounds from images and labels. Without manually defining the background texts, our learned prompts would properly describe the backgrounds associated with target object categories. Under the guidance of our class-associated background prompts, we further suppress co-occurring backgrounds from the activation maps via *Class-associated Semantic Refinement*. With the above-introduced learning strategy, we will be able to enhance the semantic matching between object regions and the associated text labels, resulting in desired activation maps for segmentation purposes. The proposed *SemPLeS* framework achieves SOTA performance on the standard WSSS benchmarks, PASCAL VOC and MS COCO. Moreover, qualitative experiments demonstrate that our learned prompts would show better interpretability than manual prompts.

In summary, our contributions are three-fold:

- We propose a ***Sem**antic **P**rompt **L**earning for WSSS (**SemPLeS**)* framework, which performs weakly-supervised semantic segmentation via enforcing semantic alignment between the object categories and the associated image regions.

- In **SemPLeS**, we present *Contrastive Prompt Learning* to learn prompts that can capture co-occurring background regions. With no need to manually define background texts, our learned prompts would represent backgrounds associated with the object categories.

- Under the guidance of the derived prompts, our *Class-associated Semantic Refinement* learns to suppress co-occurring backgrounds while enhancing the semantic matching between object regions and the associated class labels, resulting in precise pseudo masks for WSSS.

## 2 RELATED WORKS

### 2.1 WEAKLY-SUPERVISED SEMANTIC SEGMENTATION (WSSS)

The existing WSSS approaches typically follow a three-stage learning process. Firstly, the image-level labels are utilized as supervision to generate initial Class Activation Maps (CAMs) (Zhou et al. (2016); Selvaraju et al. (2017)). Secondly, the initial CAMs are refined by using dense CRF (Krähenbühl & Koltun (2011)) or pixel affinity-based methods (Ahn & Kwak (2018); Ahn et al. (2019)) to obtain pseudo masks. Lastly, the refined pseudo masks are further used as ground truth to train a segmentation network. Among all the stages, producing precise CAMs is the main focus of WSSS, and various approaches have been proposed to improve the quality of CAMs to benefit the learning of the segmentation networks (Fan et al. (2020); Chen et al. (2022a;b); Xie et al. (2022b); Du et al. (2022); Jiang et al. (2022); Li et al. (2022b); Lee et al. (2022); Yoon et al. (2022); Wu et al. (2022); Li et al. (2022c)). On the other hand, there are end-to-end WSSS works without the need for multiple training steps (Ru et al. (2023); Wu et al. (2024)), but their performances are worse than standard 3-stage methods. With the rapid development and the success of vision transformers (Dosovitskiy et al. (2021)), recent approaches (Ru et al. (2022); Xu et al. (2022a); Rossetti et al. (2022); Ru et al. (2023); Cheng et al. (2023); Xu et al. (2023b); Peng et al. (2023); Zhu et al. (2023)) generate finer activation maps based on the patch-level affinity learned from the attention layers. In general, most WSSS methods take class labels as discriminative supervision to generate CAMs without considering the textual meaning of class names. Instead, our method exploits vision-language models to learn class-associated semantic knowledge for segmentation.

### 2.2 CLIP FOR SEMANTIC SEGMENTATION

Recently, the Contrastive Language-Image Pretraining (CLIP) model (Radford et al. (2021)) has been adopted in semantic segmentation tasks thanks to the generalized knowledge learned from a large corpus of image-text pairs. Given the generalization capability, several zero-shot approaches (Li et al. (2022a); Lüddecke & Ecker (2022); Ding et al. (2022); Rao et al. (2022); Xu et al. (2022b); Ghiasi et al. (2022); Liang et al. (2023); Xu et al. (2023c;a)) exploit CLIP to segment the classes which are unseen during training. However, these methods still require mask annotations during training, causing additional manual effort. CLIP has also been adopted to improve unsupervised methods (Zhou et al. (2022a); Shin et al. (2022); He et al. (2023)). While minimizing the annotation effort, the segmentation performance is still unsatisfactory since there is no ground truth to guide the training. Recently, CLIP has also been utilized to benefit WSSS by comprehending the meaning of class-related texts using the language model (Xie et al. (2022a); Lin et al. (2023)). However, they mainly design text prompts with manual effort. Instead, our proposed SemPLeS framework *automatically* learns the class-associated prompts embedded with semantic knowledge, producing precise pseudo masks for segmentation purposes.

### 2.3 PROMPT LEARNING

In natural language processing (NLP), prompting (Liu et al. (2023)) involves giving a text-based input such as a sentence or phrase to obtain desired responses from language models. Driven by the recent success of pre-trained vision-language models (*e.g.*, CLIP (Radford et al. (2021))), there has been an increasing interest to identify proper prompts for computer vision tasks. Early work relies on prompt engineering to identify text templates (*e.g.*, "a photo of ") describing classes of interest to obtain underlying knowledge. However, such a trial and error approach generally takes a large amount of time and effort and also requires expertise about the task. To tackle the problem, prompt learning methods (Zhou et al. (2022c;b); Jia et al. (2022)) replace the manually-defined text prompts with a set of learnable context vectors preceding the class names to automate the prompting process. Different from the above methods, our *Contrastive Prompt Learning* aims to capture class-associated semantic knowledge for segmentation purposes rather than replacing the text template like "a photo of {}" for classification purposes.

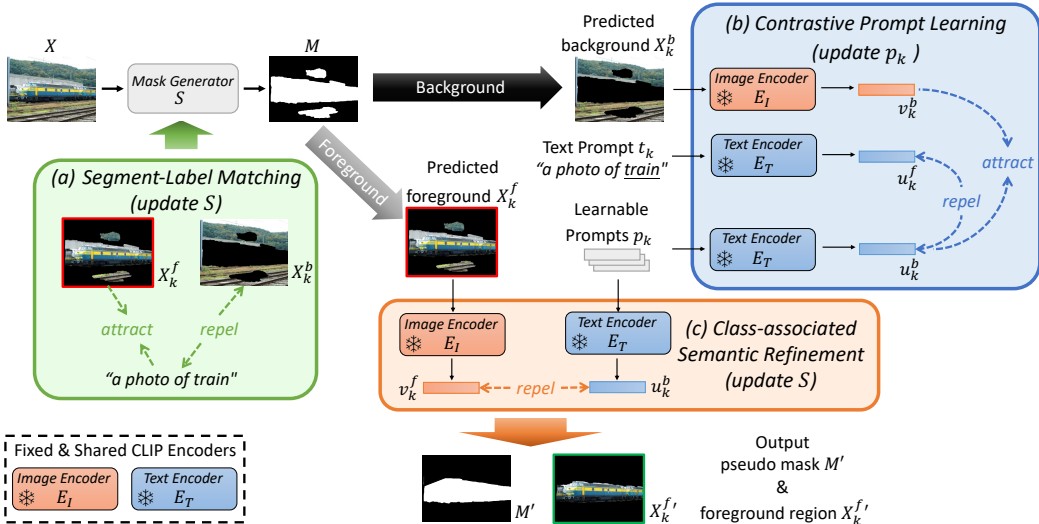

Figure 2: An overview of the proposed **SemPLeS** framework. We first introduce *Segment-Label Matching* as a warm-up stage, which leverages image-text contrastive learning to produce initial object masks $M$ from the mask generator $S$ (Sec. 3.2.1). We then propose *Contrastive Prompt Learning* (Sec. 3.2.2) and *Class-associated Semantic Refinement* (Sec. 3.2.3) to automatically learn prompts $p_k$ embedded with semantic knowledge, which can help produce refined object masks $M'$.

## 3 METHOD

### 3.1 PROBLEM FORMULATION AND MODEL OVERVIEW

We first define the problem setting and notations used in this paper. In weakly-supervised semantic segmentation (WSSS), we assume that there is a set of $N$ images $X$ with the associated image-level labels $y$, where $X \in \mathbb{R}^{H \times W \times 3}$ and $y \in \{0, 1\}^K$ is a multi-hot vector indicating the presence or absence of $K$ object categories. Without access to pixel-wise annotations, our goal is to learn precise class activation maps for segmentation purposes. To perform segmentation from image labels, we propose a novel ***Sem**antic **P**rompt **Le**arning for WSSS (**SemPLeS**)* framework to derive prompts from visual and textual representations of CLIP (Radford et al. (2021)), which would be exploited to enhance the semantic alignment between the segmented regions and the target object categories.

As shown in Figure 2, our framework takes *Segment-Label Matching* as a warm-up stage, leveraging image-text contrastive learning to produce initial object masks $M$ from our mask generator $S$ (Sec. 3.2.1). To suppress falsely activated backgrounds in such masks (*e.g.*, $X_k^f$ in the red box), we present *Contrastive Prompt Learning* (Sec. 3.2.2) and *Class-associated Semantic Refinement* (Sec. 3.2.3) to tackle this problem. The former learns class-associated prompts $p_k$ to capture co-occurring backgrounds from images and labels, while the latter takes the derived prompts to disregard co-occurring backgrounds from the object masks (*e.g.*, $X_k^{f'}$ in the green box). By jointly enforcing vision-language matching and suppression objectives, our proposed framework would enhance the semantic alignment between object regions and the associated text labels, resulting in precise activation maps for the objects of interest.

### 3.2 SEMANTIC PROMPT LEARNING FOR WSSS (SEMPLES)

#### 3.2.1 SEGMENT-LABEL MATCHING

Given an input image $X$, our mask generator $S$ is designed to produce soft foreground masks $M = S(X)$ for target object categories. Since pixel-wise annotations are not available, we choose to leverage vision-language models to guide the learning of our mask generators from image-level supervision. To be more precise, we exploit the joint latent space for images and texts from CLIP to match the object regions and the associated text labels. To achieve this, an image-text triplet (*i.e.*,

foreground-background-text) would be formulated to perform contrastive learning, as illustrated in Figure 2 (a). For the $k$th ground truth category which presents in the input image $X$ (*i.e.*, $y_k = 1$), we derive the foreground image $X_k^f = M_k \cdot X$ by applying the $k$th predicted mask $M_k$ to the original image $X$. Similarly, we reverse the predicted mask to obtain the background regions $X_k^b = (1 - M_k) \cdot X$. As for the text input $t_k$, we adopt the common prompt template "a photo of {}" filled with the $k$th class name to describe the category of interest. With the triplet $[X_k^f, X_k^b, t_k]$ serving as the input of the image encoder $E_I$ and text encoder $E_T$ pre-trained by CLIP, we perform image-text contrastive learning to maximize the cosine similarity between $X_k^f$ and $t_k$ for the foreground, while the similarity of $X_k^b$ and $t_k$ would be minimized to repel the background. Therefore, our matching loss $L_{match}$ would be formulated as follows:

$$L_{match} = \mathbb{E}_X - \left[ log(sim(v_k^f, u_k^f)) + \lambda_b \cdot log(1 - sim(v_k^b, u_k^f)) \right],$$

$$\text{where} \quad v_k^f = E_I(X_k^f), \quad v_k^b = E_I(X_k^b) \quad \text{and} \quad u_k^f = E_T(t_k).$$

(1)

Here, $\lambda_b$ is the loss weight for repelling backgrounds and $sim$ refers to cosine similarity. Note that we keep the image encoder $E_I$ and the text encoder $E_T$ frozen during training and preserve the latent space learned from CLIP to avoid potential overfitting. With the above *Segment-Label Matching* step, our mask generator $S$ is encouraged to segment object regions that align with the associated text labels. However, as noted above, such masks learned from image-level supervision are still coarse, and may falsely include co-occurring backgrounds associated with certain object categories. Therefore, the above image-text matching is not sufficient for segmentation.

### 3.2.2 CONTRASTIVE PROMPT LEARNING

To address the coarse mask issues, the previous language-guided approach (Xie et al. (2022a)) exploits vision-language models to refine the masks with manual prompting techniques. However, these methods require additional prompt engineering efforts with human knowledge involved. Moreover, manual prompting may not be able to fully exploit vision-language representation space. To tackle these problems, we propose *Contrastive Prompt Learning* (Figure 2 (b)) to learn prompts embedded with semantic knowledge from vision-language models, facilitating the following object mask refinement. Different from the previous work, we employ a sequence of learnable prompts $p_k$ as the input of the text encoder $E_T$ to describe backgrounds for the $kth$ category. Specifically, to align the prompts $p_k$ with the background image $X_k^b$, we maximize the similarity of their representations in the latent space by proposing $L_{prompt}^I$. On the other hand, to avoid describing the foreground objects, we encourage the similarity between $u_k^b$ and $u_k^f$ to be low with the proposed $L_{prompt}^T$. Thus, we propose $L_{prompt}$ as below:

$$L_{prompt} = L_{prompt}^I + \lambda_T \cdot L_{prompt}^T$$

$$= \mathbb{E}_X \left[ -log(sim(u_k^b, v_k^b)) + \lambda_T \cdot log(sim(u_k^b, u_k^f)) \right],$$

$$\text{where} \quad u_k^b = E_T(p_k), \quad v_k^b = E_I(X_k^b) \quad \text{and} \quad u_k^f = E_T(t_k).$$

(2)

Here, $\lambda_T$ is the loss weight for minimizing the similarities to the object categories. Note that the mask generator $S$ is fixed and $p_k$ is the only trainable part in this step. Once the above learning is complete, our prompts $p_k$ can represent backgrounds for the $k$th categories. It is worth noting that, our *Contrastive Prompt Learning* aims to capture class-associated backgrounds for segmentation purposes rather than replacing the text template like "a photo of {}" for classification purposes.

### 3.2.3 CLASS-ASSOCIATED SEMANTIC REFINEMENT

Finally, to suppress co-occurring background regions from the object mask $M$, our *SemPLeS* framework exploits the previously derived background prompts $p_k$ to perform *Class-associated Semantic Refinement* (Figure 2 (c)). More specifically, we encourage our mask generator $S$ to produce refined masks $M'$ by excluding the semantic knowledge embedded in the background prompts $p_k$, while the objectives introduced in 3.2.1 are retained to match the class labels. Hence, the refinement loss $L_{refine}$ and the total loss function $L_{total}$ are defined as follows:

$$L_{total} = L_{match} + \lambda \cdot L_{refine}, \quad \text{where} \quad L_{refine} = \mathbb{E}_X \left[ -log(1 - sim(v_k^f, u_k^b)) \right]. \quad (3)$$

Here, $\lambda$ is the weight for the refinement loss. It can be seen that, with the derived background prompts $p_k$ (fixed in this step) and the introduced refinement loss $L_{refine}$, the class-associated background regions would be suppressed from the foreground mask $M$, preventing possible false activation. More importantly, by jointly applying the matching and refinement objectives with image-level supervision, our *Semantic Prompt Learning for WSSS* framework advances vision-language learning to enhance the semantic alignment between the segmented regions and the target object categories, resulting in compact and complete object masks $M'$ desired for segmentation in a weakly-supervised fashion, where $M'$ can be adopted in the WSSS pipeline to obtain the final segmentation results.

## 4 EXPERIMENTS

### 4.1 DATASETS, IMPLEMENTATION, AND EVALUATION METRIC

We train and validate our proposed framework on the standard benchmark semantic segmentation datasets, PASCAL VOC 2012 (Everingham et al. (2010)) and MS COCO 2014 (Lin et al. (2014)). The PASCAL VOC 2012 dataset contains 20 object categories along with a background category. The original training, validation, and testing set consists of 1464, 1449, and 1456 images, respectively, with only image-level labels for training. The testing set results are obtained from the official evaluation website. As for the MS COCO 2014 dataset, the training and validation set contains 82081 and 40137 images from 80 object categories, respectively. We adopt the recent SOTA Transformer-based backbone model Zhu et al. (2023) for our mask generator $S$, and perform online retraining (Zhu et al. (2023)) to obtain the final segmentation model. For CLIP (Radford et al. (2021)), we use ViT-B/32 (Dosovitskiy et al. (2021)) as the image encoder. The learnable prompts are randomly initialized with the sequence length $K = 30$, and the prompt embedding dimension $D = 512$. Following the common protocol in previous works (Xie et al. (2022a)), we train the proposed framework with an augmented training set of 10582 images (e.g., flipped, cropped, etc.). The mean Intersection over Union (mIoU) is used as the evaluation metric for all experiments.

### 4.2 QUANTITATIVE COMPARISONS

To evaluate our proposed SemPLeS framework, we first compare the accuracy of the initial activation maps and also the produced pseudo masks (Seed) with previous works, as shown in Table 1. From the results in this table, our method achieves the best performance compared with existing weakly-supervised segmentation works. Specifically, our initial activation maps (Seed) achieve 68.7% while the pseudo masks (Mask) report 78.3% in mIoU. This verifies that, by advancing vision-language learning on the pre-trained CLIP model, our proposed *SemPLeS* framework successfully generates pixel-wise predictions from image-level labels, which would benefit the learning of the following segmentation model.

Table 1: Quantitative comparisons of the initially generated CAMs (Seed) and the pseudo masks on PASCAL VOC 2012 training set. The best and second results are in **bold** and underline, respectively. $^\dagger$: reproduced from the public codes.

| | Seed | Mask |
|---|---|---|
| CLIMS $_{CVPR'22}$ (Xie et al. (2022a)) | 57.5 | 72.8 |
| MCTformer $_{CVPR'22}$ (Xu et al. (2022a)) | 61.7 | 69.1 |
| Spatial-BCE $_{ECCV'22}$ (Wu et al. (2022)) | 65.3 | 66.3 |
| ViT-PCM $_{ECCV'22}$ (Rossetti et al. (2022)) | 67.1 | 71.4 |
| ESOL $_{NeurIPS'22}$ (Li et al. (2022c)) | 53.6 | 68.7 |
| ToCo $_{CVPR'23}$ (Ru et al. (2023)) | - | 72.2 |
| CLIP-ES $_{CVPR'23}$ (Lin et al. (2023)) | **70.8** | 75.0 |
| USAGE $_{ICCV'23}$ (Peng et al. (2023)) | 67.7 | 72.8 |
| WeakTr $_{arXiv'23}$ (Zhu et al. (2023))$^\dagger$ | 65.9 | 74.2 |
| **SemPLeS (Ours)** | 68.7 | **78.3** |

For the final segmentation results, our *SemPLeS* achieves the best performance and reports 74.2% and 74.8% mIoU on the validation and testing set, respectively, as shown in Table 2. Our method improves the strong baseline, WeakTr (Zhu et al. (2023)), by 1.0% and 0.8% mIoU on the validation and testing set, respectively. Moreover, our results are only 5.5% and 4.8% lower than our upper bound, which is the fully-supervised method trained with densely-annotated labels, on the validation and testing set, respectively. Similarly, our *SemPLeS* performs favorably and achieves 44.9% mIoU on MSCOCO, as shown in Table 3. The above results verify that our method is effective in performing segmentation in a weakly-supervised fashion.

Table 2: Quantitative comparisons for recent methods on PASCAL VOC 2012 *val* and *test* sets. "Sup." denotes supervision type. $\mathcal{F}$ denotes full supervision. $\mathcal{I}$ denotes image-level supervision. "Seg. Model" denotes the segmentation network. "DL1", "DL2", and "DL3" denote DeepLabV1, V2, and V3+, respectively. "ResX" and "WResX" denote ResNet-X and WideResNet-X, respectively. "Backbone" denotes the network for CAMs generation. $^\dagger$: reproduced from the public codes, where we adopt the standard backbone for a fair comparison, instead of the one with larger training scale and resolution (Steiner et al. (2022)) as shown in the paper-with-code website.

| Sup. | Method | Seg. Model | Backbone | *val* | *test* |
|---|---|---|---|---|---|
| | ***Full supervision*** | | | | |
| | DeepLabV2 $_{TPAMI'18}$ (Chen et al. (2017a)) | DL2-Res101 | - | 77.6 | 79.7 |
| $\mathcal{F}$ | WResNet38 $_{PR'19}$ (Wu et al. (2019)) | DL1-WRes38 | - | 80.8 | 82.5 |
| | Segmenter $_{ICCV'21}$ (Strudel et al. (2021)) | Seg-DeiT-S | - | 79.7 | 79.6 |
| | ***Image-level supervision only*** | | | | |
| | CLIMS $_{CVPR'22}$ (Xie et al. (2022a)) | DL2-Res101 | Res50 | 69.3 | 68.7 |
| | MCTformer $_{CVPR'22}$ (Xu et al. (2022a)) | DL1-WRes38 | DeiT-S | 71.9 | 71.6 |
| | ESOL $_{NeurIPS'22}$ (Li et al. (2022c)) | DL2-Res101 | Res50 | 69.9 | 69.3 |
| | ToCo $_{CVPR'23}$ (Ru et al. (2023)) | DeiT-B | DeiT-B | 69.8 | 70.5 |
| | CLIP-ES $_{CVPR'23}$ (Lin et al. (2023)) | DL2-Res101 | ViT-B/16 | 71.1 | 71.4 |
| $\mathcal{I}$ | OCR $_{CVPR'23}$ (Cheng et al. (2023)) | DL1-WRes38 | DeiT-S | 72.7 | 72.0 |
| | MMCST $_{CVPR'23}$ (Xu et al. (2023b)) | DL1-WRes38 | ViT-B | 72.2 | 72.2 |
| | BECO $_{CVPR'23}$ (Rong et al. (2023)) | DL3-Res101 | Res101 | 72.1 | 71.8 |
| | ACR $_{CVPR'23}$ (Kweon et al. (2023)) | DL1-WRes38 | WRes38 | 71.9 | 71.9 |
| | USAGE $_{ICCV'23}$ (Peng et al. (2023)) | DL1-WRes38 | DeiT-S | 71.9 | 72.8 |
| | FPR $_{ICCV'23}$ (Chen et al. (2023)) | DL2-Res101 | Res50 | 70.3 | 70.1 |
| | Mat-Label $_{ICCV'23}$ (Wang et al. (2023)) | DL2-Res101 | Res50 | 73.0 | 72.7 |
| | WeakTr $_{arXiv'23}$ (Zhu et al. (2023))$^\dagger$ | Seg-DeiT-S | DeiT-S | 73.2 | 74.0 |
| | MCC $_{WACV'24}$ (Wu et al. (2024)) | DeiT-B | DeiT-B | 70.3 | 71.2 |
| | **SemPLeS (Ours)** | Seg-DeiT-S | DeiT-S | **74.2** | **74.8** |

## 4.3 QUALITATIVE COMPARISONS

In addition to quantitative results, we also provide qualitative comparisons of initially generated CAMs as shown in Figure 3, and our method shows more accurate activation maps compared with CLIMS (Xie et al. (2022a)) and WeakTr (Zhu et al. (2023)). This validates that, by advancing image-text contrastive learning with learnable prompts, our *SemPLeS* would enhance the alignment between the segment regions and the target object categories. In Figure 4, we also visualize the corresponding regions of our learnable background prompts by calculating the similarities to image patches with the text and image encoders of CLIP. We see that the manual prompts defined in

Table 3: Quantitative comparisons of recent methods on MS COCO 2014 validation set. $^\dagger$: reproduced from the public codes.

| Method | Val |
|---|---|
| MCTformer $_{CVPR'22}$ (Xu et al. (2022a)) | 42.0 |
| Spatial-BCE $_{ECCV'22}$ (Wu et al. (2022)) | 35.2 |
| ESOL $_{NeurIPS'22}$ (Li et al. (2022c)) | 42.6 |
| ToCo $_{CVPR'23}$ (Ru et al. (2023)) | 41.3 |
| OCR $_{CVPR'23}$ (Cheng et al. (2023)) | 42.5 |
| USAGE $_{ICCV'23}$ (Peng et al. (2023)) | 42.7 |
| WeakTr $_{arXiv'23}$ (Zhu et al. (2023))$^\dagger$ | 44.4 |
| **SemPLeS (Ours)** | **44.9** |

CLIMS (Xie et al. (2022a)) may falsely highlight the foreground objects due to high co-occurrence when pre-training CLIP. Furthermore, such manual prompts are limited to specific categories, and therefore the corresponding backgrounds even do not present in the images (*e.g.*, the prompt "a photo of track" falsely indicates the background regions of the first example). In contrast, our learnable prompts successfully highlight all the background regions associated with each object category, demonstrating the effectiveness of our *Contrastive Prompt Learning*. It is worth noting that our class-associated prompts can also learn class-agnostic knowledge, such as the *grass* and *tree* region in both the first (cow) and third (dog) image in Figure 4.

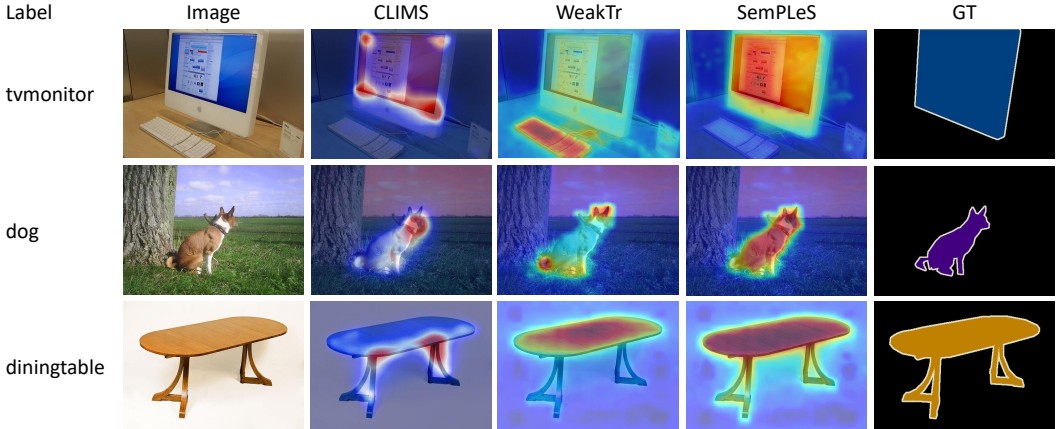

Figure 3: Qualitative comparisons of the initial CAMs on PASCAL VOC 2012. Note that our *SemPLeS* successfully suppresses the backgrounds and achieves better results than WeakTr and CLIMS, which relies on *manually-defined* background prompts with the heuristic human effort.

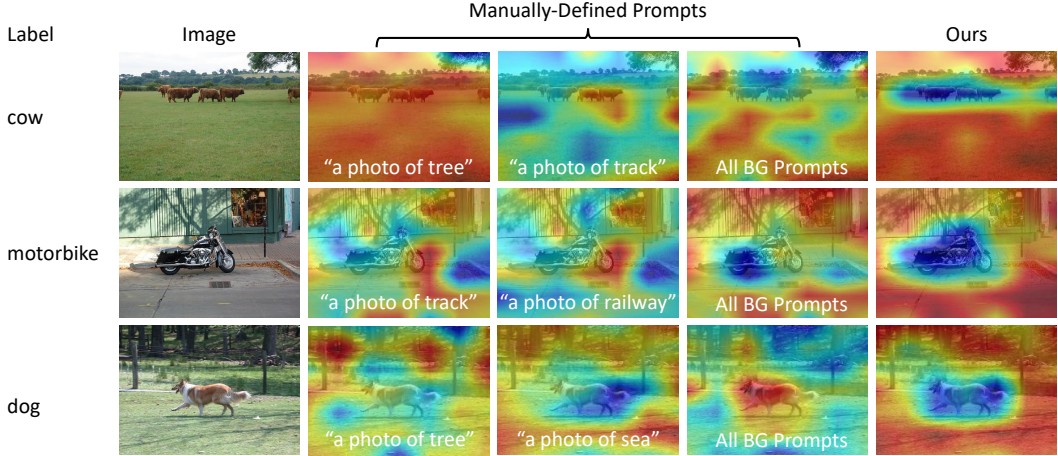

Figure 4: Visualization of manually-defined/learned background prompts and the corresponding image regions. For manual prompts (determined in CLIMS, we select those with the highest, lowest similar, and the aggregated one ("All BG Prompts") from the second to fourth columns, respectively.

## 4.4 ABLATION STUDIES

**Loss Component Analysis.** To analyze the importance of the introduced loss functions, we conduct both quantitative and qualitative ablation studies, as shown in Table 4 and Figure 5. In Table 4, we see that applying only the matching loss $L_{match}$ would result in 67.6% mIoU. When learning our background prompts, considering only the $L_{prompt}^T$ to repel the text labels may result in trivial solutions with little im-

Table 4: Quantitative ablation studies of the loss functions on PASCAL VOC 2012.

| $L_{match}$ | $L_{prompt}^T$ | $L_{prompt}^I$ | $L_{refine}$ | mIoU(%) |
|:---:|:---:|:---:|:---:|:---:|
| ✓ | | | | 67.6 |
| ✓ | ✓ | | ✓ | 67.7 |
| ✓ | | ✓ | ✓ | 67.9 |
| ✓ | ✓ | ✓ | ✓ | **68.7** |

provement. On the other hand, if only $L_{prompt}^I$ is enforced to achieve alignment with the background images, the prompts are still likely to learn the semantics of the foreground object categories instead of the background, resulting in 67.9% mIoU. Finally, when $L_{prompt}^I$ and $L_{prompt}^T$ are jointly applied to learn the background regions while avoid describing the foreground object categories, the mIoU would improve to 68.7%. Together with the qualitative results in Figure 5, we validate that our proposed framework SemPLeS and semantic prompt learning would prevent false activation of co-occurring backgrounds and therefore benefit segmentation in a weakly-supervised fashion.

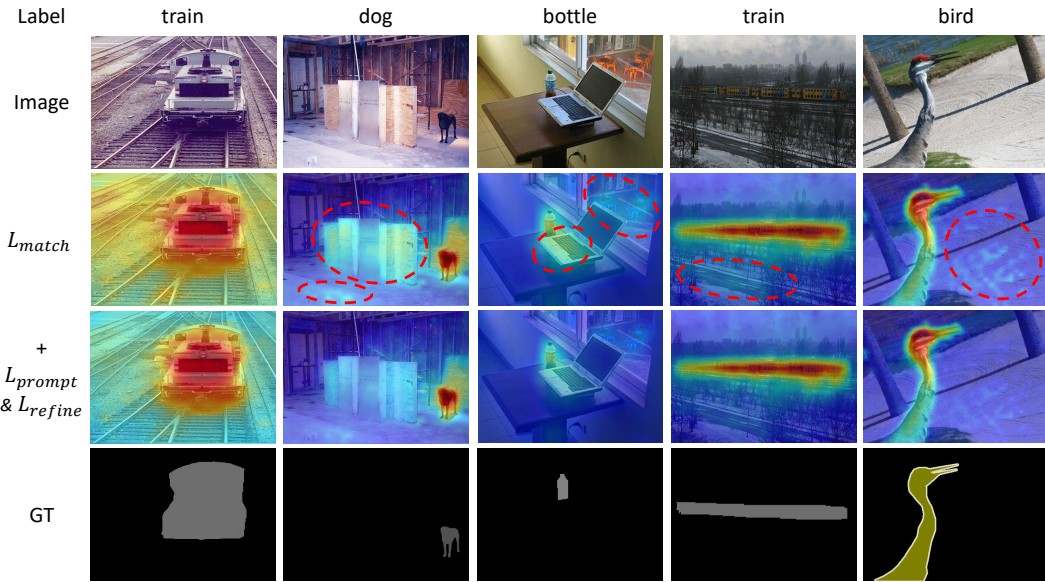

Figure 5: Qualitative ablation studies of losses. With $L_{refine}$ applied with our learned prompts, our *SemPLeS* successfully suppresses class-associated backgrounds from class activation maps.

**Loss Weight Analysis.** In Table 5, we perform hyper-parameter analysis on the refinement loss weight $\lambda$ using PASCAL VOC 2012. $\lambda = 0$ means only $L_{match}$ is used. When varying $\lambda$ from 0 to 0.1, the mIoU gradually increases from 67.6% to 68.7% when $\lambda = 0.05$, and then drops to 67.8% when $\lambda = 0.1$. Hence, we set $\lambda = 0.05$ by default.

Table 5: Loss weight analysis.

| $\lambda$ | 0 | 0.01 | 0.02 | 0.05 | 0.1 |
|---|---|---|---|---|---|
| mIoU | 67.6 | 68.3 | 68.4 | **68.7** | 67.8 |

**Learning Scheme Variants.** In Table 6, we provide quantitative comparisons when using different learning schemes in our proposed *SemPLeS* framework using PASCAL VOC 2012. We see that, jointly training all the learning steps (i.e., a, b, and c in Figure 2) in Figure 2 would result in inferior performance of 65.9%. This is because that, in the beginning of training, the prompts are not well learned and therefore would *not* be desired for our *Class-*

Table 6: Learning scheme variants analysis.

| Scheme | # of rounds | mIoU |
|---|---|---|
| joint training | - | 65.9 |
| | 1 | **68.7** |
| seq./alter. training | 2 | 68.5 |
| | 3 | 68.4 |

*associated Semantic Refinement (c)*. If we instead perform sequential training (*i.e.*, 1 round), the mIoU would improve to 68.7%. Since alternate training between *Contrastive Prompt Learning (b)* and *Class-associated Semantic Refinement (c)* results in similar performance in our *SemPLeS* framework (and also increase the training time), we choose to sequentially train the two steps once instead.

## 5 CONCLUSION

In this paper, we propose a ***Sem**antic **P**rompt **Le**arning for WSSS (**SemPLeS**)* framework, which advances vision-language learning to achieve weakly-supervised semantic segmentation (WSSS). In addition to exploiting the pre-trained CLIP model to perform *Segment-Label Matching*, we further present *Contrastive Prompt Learning* and *Class-associated Semantic Refinement* in the proposed *SemPLeS* framework to prevent false activation of image backgrounds. With no need to manually define background texts through prompt engineering, our learned prompts properly capture and suppress co-occurring backgrounds for each object category, resulting in precise activation maps for segmentation in a weakly-supervised fashion. Quantitative experiments on the segmentation benchmarks confirm the effectiveness of our proposed *SemPLeS* framework, and visualization and ablation studies are conducted to demonstrate and verify the interpretability of learned prompts.

## 6 REPRODUCIBILITY STATEMENT

The source codes are provided in the supplementary. More details are described in Sec. 4.1 and Sec. A.2.

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

# A APPENDIX

## A.1 MORE QUALITATIVE COMPARISONS

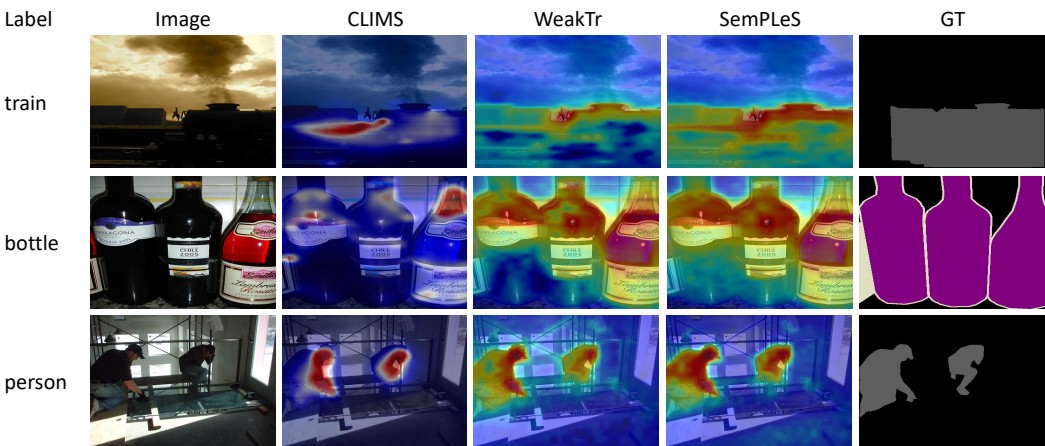

Figure 6: More qualitative comparisons of the initial CAMs on PASCAL VOC 2012.

In Figure 6, we provide qualitative results of more classes on PASCAL VOC 2012. Together with Figure 3, we see that our proposed *SemPLeS* framework would result in precise activation maps on various object categories and perform favorably compared with previous works.

## A.2 IMPLEMENTATION DETAILS

We adopt the recent SOTA Transformer-based backbone model Zhu et al. (2023) for our mask generator $S$. For CLIP (Radford et al. (2021)), we use ViT-B/32 (Dosovitskiy et al. (2021)) as the image encoder. We adopt cosine similarity where non-positive scores are clamped to a positive number $0.0001$. The default batch size is $64$. The AdamW optimizer is adopted with the cosine scheduler. We set the initial learning rate to 5e-4 and 5e-6 and train our framework for 60 epochs on PASCAL VOC 2012 and MS COCO 2014, respectively. The learnable prompts are randomly initialized with the sequence length $K = 30$, and the prompt embedding dimension $D = 512$. For loss weights, we set $\lambda_b$, $\lambda_T$ and $\lambda$ as 2.4, 0.02 and 0.05 for PASCAL VOC 2012 and 0.75, 0.01 and 0.2 for MS COCO 2014. All models are implemented in PyTorch and trained on NVIDIA V100 GPUs with 32 GB memory. We follow Zhu et al. (2023) and perform online retraining to obtain the final segmentation model.

## A.3 COMPARISON WITH CLIP-BASED WSSS WORKS.

This paper tackles the WSSS problem under the constraint of using only class labels during training, excluding the use of mask annotations. In light of this restriction, CLIP emerges as a favorable option for external knowledge incorporation due to its training without reliance on mask annotations. Notably, CLIP is designed for general purposes rather than being tailored solely for segmentation tasks. A recent trend involves leveraging knowledge within the CLIP space to enhance performance in dense prediction tasks. However, there are only a few WSSS papers based on CLIP. CLIMS (Xie et al. (2022a)) refines masks by manually prompting the pretrained CLIP model. CLIP-ES (Lin et al. (2023)) extracts attention maps from the CLIP-pretrained ViT model. MMCST (Xu et al. (2023b)) utilizes text features from the CLIP-pretrained ViT model to perform their proposed contrastive loss.

Different from the above, our method learns the class-associated prompts embedded with semantic knowledge ingrained in the CLIP space, which can be used to refine masks. The quantitative comparison in Table 7 shows that our method outperforms current CLIP-based WSSS methods. Moreover, we have also built our method upon CLIMS (Xie et al. (2022a)), demonstrating that our

Table 7: Quantitative comparisons for recent CLIP-based WSSS methods on PASCAL VOC 2012 *val* and *test* sets. "Seg. Model" denotes the segmentation network. "DL1" and "DL2" denote DeepLabV1 and V2, respectively. "ResX" and "WResX" denote ResNet-X and WideResNet-X, respectively. "Backbone" denotes the network for CAMs generation. [††]: We use the better version provided by the official GitHub repository.

| Method | Seg. Model | Backbone | *val* | *test* |
|---|---|---|---|---|
| CLIMS [CVPR'22] (Xie et al. (2022a))[††] | DL2-Res101 | Res50 | 70.4 | 70.0 |
| CLIP-ES [CVPR'23] (Lin et al. (2023)) | DL2-Res101 | ViT-B | 71.1 | 71.4 |
| MMCST [CVPR'23] (Xu et al. (2023b)) | DL1-WRes38 | ViT-B | 72.2 | 72.2 |
| **SemPLeS (Ours) with CLIMS**[††] | DL2-Res101 | Res50 | **71.0** | **71.5** |
| **SemPLeS (Ours) with WeakTr** | Seg-DeiT-S | DeiT-S | **74.2** | **74.8** |

proposed SemPLeS exhibits effectiveness with both a CNN-based mask generator, such as CLIMS, and a Transformer-based mask generator, like WeakTr, and improves both CLIMS and WeakTr, respectively. This also implies that our proposed SemPLeS can work with different segmentation models (e.g., CLIMS and WeakTr), and improve both performances, respectively. It is also worth noting that there is no additional complexity burden for the training pipeline. The training pipeline of our backbone, WeakTr, has two stages including CAM generation for pseudo-label, and online re-training for final segmentation. The whole CLIP-ES (Lin et al. (2023)) pipeline also has two stages, including CAM generation with refinement for pseudo-label, and segmentation model training with their proposed confidence-guided loss. Therefore, replacing the mask generator with CLIP-ES will not reduce the training pipeline complexity.

On the other hand, most CLIP-based papers are related to the open-vocabulary setting with the requirement of mask annotation training, which is beyond the scope of this paper. For example, OVSeg (Liang et al. (2023)) is trained with COCO-Stuff segmentation annotations, which violates the WSSS protocol. MaskCLIP+ (Zhou et al. (2022a)) adopts the transductive setting, so the testing data is seen during training, which also violates the WSSS protocol. Nonetheless, there is an intriguing avenue for exploration in integrating certain concepts from OVSeg (e.g., learnable mask prompt) or MaskCLIP (e.g., prompt denoising) into our SemPLeS framework. We consider this as a potential direction for our future work.

## A.4 More Discussions

**Comparison with MARS (Jo et al. (2023)).** Recently, MARS (Jo et al. (2023)) has been published and shows significantly better performance than all the other WSSS works including ours. We conjecture the reason for the performance is because MARS builds upon a strong baseline RS-EPM (Jo et al. (2022)) with a strong backbone DeepLabv3+ (ResNet101), and it also proposes to exploit an additional strong Unsupervised Semantic Segmentation model, STEGO (Hamilton et al. (2022)) with ViT-B, to remove biased objects and achieve the current SOTA for WSSS. This method is interesting and we envision its potential integration with our automatic prompt learning method for further improvement. We leave this prospect as part of our future work.

**Comparison with SAM (Kirillov et al. (2023)).** SAM (Kirillov et al. (2023)) has been recently proposed as a foundation segmentation model which can be combined with open-vocabulary semantic segmentation methods, leading to enhanced performance. However, our paper focuses on the challenging WSSS problem, specifically in the scenario where only class labels are available, without the use of semantic or class-agnostic mask annotations during training. SAM's training on the SA-1B dataset including over 1 billion mask annotations, places it beyond the scope of this work. Nonetheless, we find it intriguing to explore how our SemPLeS framework could enhance SAM's semantic understanding capabilities, offering valuable insights into WSSS under different constraints. WSSS with SAM is an under-explored research avenue, involving class-agnostic mask annotation and image-level labels, and we consider this as a potential direction for our future work.

**Comparison with Panoptic Segmentation.** To our best understanding, although there are some weakly-supervised panoptic segmentation papers like WSSPS (box-supervised) Li et al. (2018), PSPS (point-supervised) Fan et al. (2022), and Panoptic-FCN (point-supervised) Li et al. (2022d), there is no panoptic segmentation work with class supervision. It would be an interesting research direction to explore, and we see it as a potential future work.

**Comparison with SPML (Ke et al. (2021)).** SPML (Ke et al. (2021)) and our method both conduct contrastive learning, but they focus on pixel-segment contrastive learning while we focus on image-text and text-text contrastive learning.

