# OpenReview forum: "SemPLeS: Semantic Prompt Learning for Weakly-Supervised Semantic Segmentation"
_ICLR.cc/2024/Conference — Submitted to ICLR 2024_

### Official Review · Reviewer_LBiv · 2023-10-29

**Soundness:** 4 excellent
**Presentation:** 4 excellent
**Contribution:** 3 good
**Rating:** 3
**Confidence:** 5

**Summary:**

This paper proposed a novel method for weakly-supervised semantic segmentation (WSSS) by exploiting the large vision-language model, CLIP. By adopting a pre-trained WSSS model as a mask generator, they update the parameters of the generator using contrastive learning between the image-text triplet. After that, the learnable prompts are trained by contrastive prompt learning. Then, the mask generator is further updated using the proposed class-associated semantic refinement. By leveraging the CLIP-based image-text aligning, they improve the strong baseline, WeakTr, and achieve state-of-the-art performances on VOC 2012 and COCO 2014 benchmarks.

**Strengths:**

1. The paper is well-written, and the description of the proposed methods is clear with well-illustrated figures.

2. The proposed method (segment-label matching, contrastive prompt learning, class-associated semantic refinement) is intuitive and convincing.

3. The experiment is somewhat complete and the proposed methods are well-ablated.

**Weaknesses:**

## 1. complex training pipeline.

As I understand, this method is a refinement method for existing WSSS methods using knowledge of CLIP.
Namely, this paper adopts WeakTr as a strong baseline WSSS method and refines it using CLIP-based contrastive learning.
From a from-scratch training perspective, WeakTr requires a two-step training pipeline (CAM generation and online retraining) and SemPLeS requires a three-step training pipeline (segment-label matching, contrastive prompt learning, and class-associated semantic refinement).

I wonder if the mask generator could be replaced by attention maps of CLIP or activation maps from CLIP (as done in CLIP-ES).

## 2. dependency of the mask generator.
I have a concern that the performance of the proposed method may largely depend on the mask generator. If the mask generator fails to generate proper segmentation masks, the WSSS performance is largely dropped.
I guess the proposed (refinement) method can be adapted to other WSSS methods, but there is only one experiment using WeakTr.

## 3. more meaningful comparison.
The methods in Table 3 can be categorized into two groups, CLIP-based methods (CLIMS and CLIP-ES) and CLIP-free methods.
Considering that CLIMS achieved 68.7% and CLIP-ES achieved 71.4% in VOC 2012 testset, the baseline method, WeakTr, already shows a much higher performance of 74.0%.
Although SemPLeS achieved a 74.8% performance, I think this outstanding performance is mainly from WeakTr.
It would be great if there were any attempts to compare with CLIP-based methods more meaningfully (e.g., using the same seg model or the same baseline model).

Also, even though the SemPLeS used the large-scale vision-language model (CLIP) with the complex three-step refinement training, it is questionable that the 0.5% improvement in COCO 2014 validation set (Table 3) is promising.

**Questions:**

Overall, I think the proposed method is interesting and convincing.
However, I have some concerns related to the complex training pipeline and dependency of the mask generator model.

Therefore, my initial rating is weak reject (4), and I finalize the rating after rebuttals.

**Details Of Ethics Concerns:**

I conclude that there are no ethics concerns in this paper.

---

> ### Author Response · Authors · 2023-11-16
> **Rebuttal**
>
> Thank you for all the constructive comments. Here are our replies:
>
> **1. Complex training pipeline:**
>
> The whole WeakTr pipeline has two stages: 1) CAM generation for pseudo-label, and 2) online retraining for final segmentation. The whole CLIP-ES pipeline also has two stages: 1) CAM generation with refinement for pseudo-label, and 2) segmentation model training with their proposed confidence-guided loss.
> Therefore, replacing the mask generator with CLIP-ES will not reduce the training pipeline complexity.
>
> **2. Dependency of the mask generator:**
> We have also built our method upon CLIMS (we use the better version provided by the official repo). The results are as follows:
>
> |Methods			|Val|	Test|
> |---|---|---|
> |CLIMS (CVPR’22)|		70.4| 	70.0|
> |CLIP-ES (CVPR’23)|		71.1|	71.4|
> |WeakTr|			73.2|	74.0|
> |SemPLeS w/ CLIMS (Ours)|	**71.0**|	**71.5**|
> |SemPLeS w/ WeakTr (Ours)|	**74.2**|	**74.8**|
>
> The above table demonstrates that our proposed SemPLeS exhibits effectiveness with both a CNN-based mask generator, such as CLIMS, and a Transformer-based mask generator, like WeakTr, and improves both CLIMS and WeakTr, respectively.
>
> **3. More meaningful comparison:**
>
> As shown in the above table, our proposed SemPLeS can work different segmentation models (e.g., CLIMS and WeakTr), and improve both performances, respectively.
>
> COCO 2014 is a challenging dataset for the WSSS problem. The average mIoU for WSSS papers published at top conferences (e.g., CVPR, ECCV, NeurIPS, etc.) was 43.11 in 2022 and increased to 43.58 in 2023. Notably, the overall improvement in WSSS within the community over the past year is estimated at around 0.47. Hence, the achieved 0.5 improvement is a non-negligible advancement in the context of WSSS performance.

---

> > ### Author Response · Authors · 2023-11-22
> >
> > We wish to emphasize that we have diligently addressed every comment. We would also be appreciative if you could consider re-evaluating the rating of this work. Please let us know if there are any concerns or questions, and we are eager to provide further clarification or discussion.

---

### Official Review · Reviewer_ZuKn · 2023-10-31

**Soundness:** 2 fair
**Presentation:** 2 fair
**Contribution:** 2 fair
**Rating:** 5
**Confidence:** 5

**Summary:**

This paper propose a prompt-learning method to enhance weakly-supervised semantic segmentation prediction.  Existing works often utilize pre-trained CLIP models to guide the class-specific foreground mask predictions, however, they often fail to separate co-occurring background categories from foreground (e.g. train vs. railroad, horse vs. grass).  Prior works address such issues by manually design background prompts for each category, hoping to refine the predicted pseudo mask.  Such methods require human efforts to manually annotate the background prompts.  This paper propose a stage-learning technique, by first training object mask predictors and then background prompts with image-text contrastive learning.  This paper demonstrates SOTA performance on standard benchmakrs, such as VOC and MSCOCO.

**Strengths:**

1. The idea of separating co-occurring background from foreground for each class makes a lot of sense.  In particular, the examples of the train and co-occurring railroad is convincing.

2. The results are SOTA on both benchmarks.

3. The paper is well written and easy to follow.

**Weaknesses:**

1. There are some citations missing:
    * ***intra-class foreground-background discrimination***: Learning Integral Objects With Intra-Class Discriminator for Weakly-Supervised Semantic Segmentation.  Fan et al. CVPR 2020.
    * ***pixel-wise contrastive learning for WSSS***: Universal Weakly Supervised Segmentation by Pixel-to-Segment Contrastive Learning. Ke et al. ICLR 2021.

2. The efficacy of class-specific background prompt is not clear.  In Table 4, training with $L_{prompt}^f$ seems to be more effective than $L_{prompt}^b$ and $L_{refine}$.

3. Segment Anything (SAM) is a strong framework for mask proposals.  Without fine-tuning on VOC / MSCOCO, I believe SAM can still provide high-quality segmentation on out-of-distribution imagery.  Reasonable baselines are: 1) classifying mask proposals with SAM using OVSeg[1]/CLIP/MaskCLIP+[2] features, and 2) instead of learning background prompts, one can learn to refine per-category mask proposals using dense segmentations derived from SAM.  We can vote the foreground confidence within each SAM segment (very like refine binary masks with CRF).  However, this paper does not provide any comparison to SAM (see questions for details).


[1] Open-Vocabulary Semantic Segmentation with Mask-adapted CLIP. Liang et al.  CVPR 2023.
[2] Extract Free Dense Labels from CLIP. Zhou et al. ECCV 2022.

**Questions:**

1. My first concern is the idea of using CLIP to guide mask predictions.  CLIP is notoriously know to perform poorly on segmentation.  May works have been proposed to address such issues (e.g. OVSeg[1] and MaskCLILP+[2]).  My question is why the authors choose to use CLIP but not MaskClip+ features?  In fact, probably we don't even need the proposes background prompt learning when using mask-sensitive CLIP features. The authors should provide analysis on OVSeg/MaskCLIP+ features.

2. I'm not sure how necessary it is to train the Mask Predictor. SAM is an existing strong segmentation framework.  Why not apply OVSeg[2] (with intra-class background prompt tuning) on mask proposals from SAM?  Or even refine the mask predictions with SAM.

3. In Table 4, what's the performance if you train with only $L_{match}$ and $L_{prompt}^f$?  It seems to me that training with $L_{prompt}^f$ brings the most performance gain, meaning the background prompts might not be as effective as the paper claims.

---

> ### Author Response · Authors · 2023-11-16
> **Rebuttal**
>
> Thank you for all the constructive comments. Here are our replies:
>
> **1. Citation missing:**
>
> Thanks for sharing. We will cite both ICD [1] and SPML [2] in our revised paper.
> Our difference from the above two papers:
> * ICD [1] and our method both aim to separate the foreground and the background, but they achieve this by learning intra-class boundaries while we learn the class-associated prompts embedded with semantic knowledge.
> * SPML [2] and our method both conduct contrastive learning, but they focus on pixel-segment contrastive learning while we focus on image-text and text-text contrastive learning.
>
> **2. Confusion about class-specific background prompts:**
> The whole $L_{prompt}$ in Eq. (2) is used to learn class-associated background prompts, including $L^b_{prompt}$ and $L^f_{prompt}$:
> * $L^b_{prompt}$ aims to attract class-associated background prompts to predicted background images
> * $L^f_{prompt}$ aims to repel class-associated background prompts from foreground class labels
>
> After the class-associated background prompts are learned, we use the learned prompts to refine the mask with $L_{refine}$. Therefore, without $L_{refine}$, we cannot know how $L^b_{prompt}$ and $L^f_{prompt}$ affect the final mIoU performance. That is to say, training with only $L_{match}$ and $L^f_{prompt}$ will provide the same results as training with $L_{match}$.
>
> **3. Comparison w/ SAM & Why not SAM:**
>
> Thanks for the insightful comments. We acknowledge the effectiveness of SAM [5] as an established segmentation model and we agree that combining SAM with open-vocabulary semantic segmentation methods, such as OVSeg, leads to enhanced performance. However, our paper focuses on the challenging WSSS problem, specifically in the scenario where only class labels are available, without the use of mask annotations during training. SAM's training on the SA-1B dataset including over 1 billion mask annotations, places it beyond the scope of the WSSS problem. Nonetheless, we find it intriguing to explore how our SemPLeS framework could enhance SAM's semantic understanding capabilities, offering valuable insights into WSSS under different constraints. We consider this as a potential direction for our future work.
>
> **4. Why CLIP instead of OVSeg/MaskCLIP+:**
>
> This paper tackles the WSSS problem under the constraint of using only class labels during training, excluding the use of mask annotations. In light of this restriction, CLIP emerges as a favorable option for external knowledge incorporation due to its training without reliance on mask annotations. Notably, CLIP is designed for general purposes rather than being tailored solely for segmentation tasks. A recent trend involves leveraging knowledge within the CLIP space to enhance performance in dense prediction tasks. We aim to harness the semantic knowledge ingrained in the CLIP space, contributing to the improvement of WSSS training.
>
> The reason why OVSeg [4] and MaskCLIP+ [3] are not better choices for this paper:
> * OVSeg (CVPR’23) [4]: it is trained w/ COCO-Stuff segmentation annotations, which violates the WSSS protocol.
> * MaskCLIP+ (ECCV’22) [3]: it uses the transductive setting, so the testing data is seen during training, which also violates the WSSS protocol.
>
> Nonetheless, there is an intriguing avenue for exploration in integrating certain concepts from OVSeg (e.g., learnable mask prompt) or MaskCLIP (e.g., prompt denoising) into our SemPLeS framework. We consider this as a potential direction for our future work.
>
> ---
> References:
> * [1] (ICD) Junsong Fan, et al. “Learning Integral Objects with Intra-Class Discriminator for Weakly-Supervised Semantic Segmentation”, CVPR, 2020.
> * [2] (SPML) Tsung-Wei Ke, et al. “Universal Weakly Supervised Segmentation by Pixel-to-Segment Contrastive Learning”, ICLR, 2021.
> * [3] (MaskCLIP) Chong Zhou, et al. “Extract Free Dense Labels from CLIP”, ECCV, 2022.
> * [4] (OVSeg) Feng Liang, et al. “Open-Vocabulary Semantic Segmentation with Mask-adapted CLIP”, CVPR, 2023.
> * [5] (SAM) Alexander Kirillov, et al. “Segment Anything”, ICCV, 2023.
>
> We will add missing citations in the revised paper.

---

> > ### Comment · Reviewer_ZuKn · 2023-11-21
> >
> > Thanks for the response.
> >
> > ---
> >
> > **Re: 2. Confusion about class-specific background prompts**
> >
> > Thanks for the explanation.  I would encourage the authors to refactor the loss notations.  It is somewhat difficult to follow how each loss term differ from others without multiple rounds of re-reading.
> >
> > ---
> >
> > **Re: 3. Comparison w/ SAM & Why not SAM**
> >
> > I disagree with the statement *"SAM's training on the SA-1B dataset including over 1 billion mask annotations, places it beyond the scope of the WSSS problem"*.  The main reason is that WSSS is intrinsically a pixel-classification task, while SAM is not trained with semantic segmentation, but general image segmentation.  In addition, the segmentation quality of SAM is much higher than most of the existing mask proposal methods, which could facilitate the research topic of WSSS.
> >
> > ---
> >
> > **Re: 4. Why CLIP instead of OVSeg/MaskCLIP+**
> >
> > Thanks for the clarification.  The response makes sense to me.

---

> ### Author Response · Authors · 2023-11-21
>
> **Re: Re: 2. Confusion about class-specific background prompts**
>
> Thanks for the valuable feedback. We will change the notation as follows:
> * $L^b_{prompt}$ $\rightarrow$ $L^I_{prompt}$: the loss is computed using class-associated background prompts and predicted background *images*.
> * $L^f_{prompt}$ $\rightarrow$  $L^T_{prompt}$: the loss is computed using class-associated background prompts and foreground class *labels*.
>
> We will revise the whole paper accordingly.
>
> **Re: Re: 3. Comparison w/ SAM & Why not SAM**
>
> Thank you for the comments.
>
> The annotation for semantic segmentation includes two aspects: *localization* and *classification*.
> Fully-supervised semantic segmentation (FSSS) relies on full mask annotation for training, with pixel-level localization and class labels. Methods without full supervision can be categorized as WSSS. However, the current convention of the WSSS community defines **Weak** only in terms of *localization*, which means that the ground truth does not contain precise pixel-level location information, but with (sparse) point-level, box-level, or image-level supervision instead. Our focus is exclusively on image-level supervision, lacking any location information.
>
> SAM, although falling under the WSSS umbrella according to the aforementioned definition due to the absence of class annotation, surpasses the focus of this study by incorporating foreground/background annotations. We are sorry for the confusion caused by our previous comments. While SAM aligns with the broader WSSS definition, it extends beyond the scope of *WSSS with image-level supervision* in this work. This work focuses on the problem with only image-level supervision without any location information.
>
>
> We appreciate the discussion prompted by your inquiries. *WSSS with SAM* is an under-explored research avenue, involving class-agnostic mask annotation and image-level labels.
> We find it intriguing to explore how our SemPLeS framework could benefit this direction, offering valuable insights into WSSS under different constraints. We see this as a prospective direction for our future work. Thank you for initiating this valuable discussion.

---

> > ### Comment · Reviewer_ZuKn · 2023-11-21
> >
> > Thanks for the response.
> >
> > ---
> >
> > **Re: Re: 2. Confusion about class-specific background prompts**
> >
> > My concern is addressed
> >
> > ---
> >
> > **Re: Re: 3. Comparison w/ SAM & Why not SAM**
> >
> > Yes! I do believe the WSSS task could be mostly solved using SAM.  In fact, there're exisitng works applying similar ideas.  For example, a recent NeuRIPS paper "Weakly-Supervised Concealed Object Segmentation with SAM-based Pseudo Labeling and Multi-scale Feature Grouping".
> >
> > My personal perspective on WSSS is that working merely on COCO/VOC dataset is obsolete, especially after the invention of SAM.  The community could instead explores what type of weakly-supervised segmentation tasks that SAM fail to solve.  I can name few of which: hierarchical segmentation and concealed object segmentation.  It'd be interesting to study weakly-supervised learning on both topics.

---

> ### Author Response · Authors · 2023-11-22
>
> Thank you for your insightful feedback.
>
> We agree that SAM reshapes the research directions in the community, especially for segmentation-related tasks, and SAM can solve various types of WSSS problems. However, *WSSS with image-level supervision* remains a formidable challenge. Recent investigations [2-5] have explored the potential of leveraging SAM for this specific problem.
> Notably, most of them integrate SAM with additional foundation models (e.g., BLIP-2 [6], Grounding-DINO [7], RAM [8]) to enhance SAM’s ability to incorporate semantic information. The results in the below table demonstrate that even with SAM, the *WSSS with image-level supervision* problem is still challenging, and our proposed SemPLeS has already outperformed several SAM-based methods.
>
> |Methods			|Val|	Test|
> |---|---|---|
> |CLIMS + SEPL (with SAM) [2]	|71.1|	      -|
> |BLIP-2 + SAM [3]			|71.1|	72.2|
> |Grounding-DINO + SAM [4]		|72.7|	72.8|
> |RAM + Grounding-DINO + SAM [4]	|74.0|	73.8|
> |Grounding-DINO(*) + SAM [5]	|77.2|	77.1|
> |SemPLeS (Ours)			|74.2|	74.8|
> (*with advanced text prompt design)
>
>
> We chose VOC and COCO since they are widely recognized benchmark datasets for this problem at the time of submission to ICLR’24. We acknowledge your suggestion to focus future research directions on tasks where SAM faces challenges or is inapplicable.
>
> In accordance with the feedback you provided, *data domain* emerges as a prominent challenge, particularly when data distributions deviate significantly from SAM's original image training data, encompassing scenarios such as concealed objects [1] and glass objects [11]. Addressing this issue requires the adoption of additional methodologies. Moreover, *model efficiency* is also another issue. SAM's application in resource-limited or time-sensitive scenarios, such as mobile applications, is hindered by its large model size. Exploring strategies to enhance SAM's efficiency without compromising accuracy represents a compelling and practical avenue for future research. Furthermore, similar to most deep models, *robustness* is another issue that SAM is facing [9-10]. We consider these directions as our future work.
>
> Thank you for continuing this valuable discussion. We hope our responses address your concern. We would also be appreciative if you could consider re-evaluating the rating of this work. Please let us know if there are any concerns or questions, and we are eager to provide further clarification or discussion.
>
> ---
> References:
> * [1] Chunming He, et al. “Weakly-Supervised Concealed Object Segmentation with SAM-based Pseudo Labeling and Multi-scale Feature Grouping”, NeurIPS, 2023.
> * [2] Tianle Chen, et al. “Segment Anything Model (SAM) Enhanced Pseudo Labels for Weakly Supervised Semantic Segmentation”, NeurIPS Workshop, 2023.
> * [3] Peng-Tao Jiang, et al. “Segment Anything is A Good Pseudo-label Generator for Weakly Supervised Semantic Segmentation”, arXiv, 2023.
> * [4] Zhaozheng Chen, et al. “Weakly-Supervised Semantic Segmentation with Image-Level Labels: from Traditional Models to Foundation Models”, arXiv, 2023.
> * [5] Weixuan Sun, et al. “An Alternative to WSSS? An Empirical Study of the Segment Anything Model (SAM) on Weakly-Supervised Semantic Segmentation Problems”, arXiv, 2023.
> * [6] Junnan Li, et al. “BLIP-2: Bootstrapping Language-Image Pre-training with Frozen Image Encoders and Large Language Models”, ICML, 2023.
> * [7] Shilong Liu, et al. “Grounding DINO: Marrying DINO with Grounded Pre-Training for Open-Set Object Detection”, arXiv, 2023.
> * [8] Youcai Zhang, et al. “Recognize Anything: A Strong Image Tagging Model”, arXiv, 2023.
> * [9] Yuqing Wang, et al. “An Empirical Study on the Robustness of the Segment Anything Model (SAM)”, arXiv, 2023.
> * [10] Yu Qiao, et al. “Robustness of SAM: Segment Anything Under Corruptions and Beyond”, arXiv, 2023.
> * [11] Dongsheng Han, et al. “Segment Anything Model (SAM) Meets Glass: Mirror and Transparent Objects Cannot Be Easily Detected”, arXiv, 2023.

---

### Official Review · Reviewer_WM2o · 2023-11-02

**Soundness:** 2 fair
**Presentation:** 2 fair
**Contribution:** 2 fair
**Rating:** 5
**Confidence:** 5

**Summary:**

The paper proposes a Semantic Prompt Learning for WSSS (SemPLeS) framework. The author proposes contrastive prompt learning to acquire class-associated background prompts and further proposes a Class-associated Semantic Refinement module to suppress erroneous associations of co-occurring backgrounds. This method shows better performance.

**Strengths:**

1. The authors propose a novel CLIP prompt method that was automatically learned rather than manually designed, which effectively promotes the alignment of the semantic space.

2. The authors provide a detailed explanation of the modulation process, demonstrating the effectiveness of learnable prompts.

3. The logic of this paper is clear and easy to read.

**Weaknesses:**

1. In Sec. 2.1, the author briefly introduces the current research status of WSSS three-stage learning. However, this method is only a research branch of WSSS, and the end-to-end method should be supplemented. Furthermore, as far as we know, the recent WSSS research is basically based on CLIP. The author's innovation lies in the non-manual design of prompts rather than establishing vision-language associations, thus there is no need to emphasize the contribution of using CLIP.

2. In Figure 2 (a), symbol abbreviations X_k^f and X_k^b are given for the foreground and background of the image, and the text prompt t_k should also be indicated in this figure.

3. In Sec. 3.2.2, it writes “learnable prompts p_k as the input of the text encoder E_t”. It is necessary to illustrate how to initialize it and its shape. The motivation for learning prompts is not new, there exist many works that learn prompts in the CLIP community, such as COOP. More differences between these methods should be discussed.

4. Missed comparison with other methods, such as the work titled "MARS: Model-agnostic Biased Object Removal without Additional Supervision for Weakly-Supervised Semantic Segmentation " from ICCV 2023. To ensure a comprehensive research evaluation and establish a fair assessment of the proposed method's performance, the authors should include extensive analysis and comparison with more SOTAs.

5. Figures 3 and 4 can be augmented by incorporating a dedicated column on the left-hand side to present image labels, rather than embedding them within the figure itself. This graphical modification not only improves visual clarity but also aligns with the model's separate processing of images and labels.

6. What does "All BG Prompts" in picture 4 mean?

7. Result of L_match + L_prompt^b + L_prompt^f  should be added in Table 4.

**Questions:**

See weakness.

---

> ### Author Response · Authors · 2023-11-16
> **Rebuttal (1/2)**
>
> Thank you for all the constructive comments. Here are our replies:
>
> **1. (i) End-to-end WSSS:**
>
> Most WSSS papers follow the standard 3-stage pipeline. There are a few end-to-end WSSS papers, but their performances are worse than standard 3-stage methods. The quantitative comparison in the below table shows that our method significantly outperforms current SOTA end-to-end methods, including ToCo [4] and MCC [14].
> We will introduce more end-to-end methods in Related Works.
>
> | Methods	| Seg. Model | Backbone | Val | Test |
> |-----------------|--------------|-------------|------|-----|
> | ToCo (CVPR’23) [4] | DeiT-B | DeiT-B | 69.8 | 70.5 |
> | MCC (WACV’24) [14] | DeiT-B | DeiT-B | 70.3 | 71.2 |
> | SemPLeS (Ours) | DeiT-S | DeiT-S | **74.2** | **74.8** |
>
> **(ii) Contribution of using CLIP:**
>
> There are not many WSSS papers based on CLIP. Most CLIP-based papers are related to the open-vocabulary setting which requires mask annotations for training. We summarize recent CLIP-based WSSS papers and explain how they exploit CLIP as follows:
> * CLIMS (CVPR. 2022) [3]: refine masks by manually prompting the pretrained CLIP model.
> * CLIP-ES (CVPR, 2023) [5]: extract attention maps from the CLIP-pretrained ViT model
> * MMCST (CVPR, 2023) [7]: use text features from the CLIP-pretrained ViT model to perform their proposed contrastive loss.
>
> Different from the above, we learn the class-associated prompts embedded with semantic knowledge, which can be used to refine masks.
> The quantitative comparison in the below table shows that our method outperforms current CLIP-based WSSS methods.
>
> |Methods			|Val|	Test|
> |---|---|---|
> |CLIMS (CVPR’22) [3]|		69.3 |	68.7|
> |CLIP-ES (CVPR’23) [5]	|71.1|	71.4|
> |MMCST (CVPR’23) [7]	|72.2|	72.2|
> |SemPLeS (Ours)		|**74.2**|	**74.8**|
>
> **2. Symbols in Figure 2 (a):**
>
> Thanks for the suggestion, and we will revise the figure as suggested.
>
> **3. (i) details of text prompt t_k (how to initialize & shape):**
>
> The learnable prompts are randomly initialized with the shape K by D, where K is prompt length, and D is the prompt embedding dimension of CLIP.
>
> **(ii) difference from current prompt learning methods:**
>
> As mentioned in Related Works, our Contrastive Prompt Learning aims to capture class-associated semantic knowledge for segmentation purposes rather than replacing the manual-defined text template like “a photo of {}” for classification purposes as in current prompt learning methods.
>
> **4. More SOTA comparison (including MARS):**
>
> We are not sure if it’s reasonable to ask for a comparison with publications where the conference dates are after the ICLR’24 submission deadline, such as ICCV 2023.
>
> Nevertheless, we list all WSSS papers published in CVPR and ICCV 2023 in the below table with the heatmap and segmentation backbones. Our method outperforms almost all other methods in the table except for MARS (ICCV’23) [10].
> MARS [10] builds upon a strong baseline RS-EPM (arXiv’ 22) [1] with a strong backbone DeepLabv3+ (ResNet101), and it also proposes to exploit an additional strong Unsupervised Semantic Segmentation model, STEGO (ICLR’ 22) [2] with ViT-B, to remove biased objects and achieve the current SOTA for WSSS. This method is interesting and we envision its potential integration with our automatic prompt learning method for further improvement. We leave this prospect as part of our future work.
>
> |Methods	|		Seg. Model|	Backbone|	Val|	Test|
> |-----------------|--------------|-------------|------|-----|
> |ToCo (CVPR’23) [4]|	DeiT-B	|	DeiT-B	|	69.3 |	68.7|
> |CLIP-ES (CVPR’23) [5]|	DL2-Res101|	ViT-B|		71.1|	71.4|
> |OCR (CVPR’23) [6]|	DL1-WRes38|	DeiT-S|		72.7	|72.0|
> |MMCST (CVPR’23) [7]|	DL1-WRes38|	ViT-B|		72.2|	72.2|
> |ACR (CVPR’23) [8]	|	DL1-WRes38|	WRes38|	71.9|	71.9|
> |BECO (CVPR’23) [9]	|	DL3+-Res101|	Res101|	72.1|	71.8|
> |MARS (ICCV’23) [10]	|	DL3+-Res101|	Res101|	***77.7***|	***77.2***|
> |USAGE (ICCV’23) [11]|	DL1-WRes38|	DeiT-S|		71.9|	72.8|
> |FPR (ICCV’23) [12]|		DL2-Res101|	Res50|		70.3	|70.1|
> |Mat-Label (ICCV’23) [13]|	DL2-Res101|	Res50|		73.0|	72.7|
> |SemPLeS (Ours)|		DeiT-S|		DeiT-S|		**74.2**	|**74.8**|
>
>
> **5. Labels presented in Figures 3 and 4:**
>
> Thanks for the suggestion, and we will revise the figure as suggested.
>
> **6. “All BG Prompts” in Figure 4:**
>
> “All BG Prompts” means the visualization is generated from ALL background prompts manually defined in CLIMS [3].
>
> **7. Table 4 clarification:**
>
> $L_{prompt}^b$ and $L_{prompt}^f$ are used to learn the background prompts, and then we use the learned prompts to refine the mask with $L_{refine}$. Therefore, without $L_{refine}$, we cannot know how $L_{prompt}^b$ and $L_{prompt}^f$ affect the final mIoU performance. Training with $L_{match}$ + $L_{prompt}^b$ + $L_{prompt}^f$ will be the same as training with $L_{match}$.

---

> > ### Author Response · Authors · 2023-11-16
> > **Rebuttal (2/2)**
> >
> > References:
> > * [1] (RS+EPM) Sanghyun Jo, et al. “RecurSeed and EdgePredictMix: Single-stage Learning is Sufficient for Weakly-Supervised Semantic Segmentation”, arXiv, 2022.
> > * [2] (STEGO) Mark Hamilton, et al. “Unsupervised Semantic Segmentation by Distilling Feature Correspondences”, ICLR, 2022.
> > * [3] (CLIMS) Jinheng Xie, et al. “Cross Language Image Matching for Weakly Supervised Semantic Segmentation”, CVPR, 2022.
> > * [4] (ToCo) Lixiang Ru, et al. “Token Contrast for Weakly-Supervised Semantic Segmentation”, CVPR, 2023.
> > * [5] (CLIP-ES) Yuqi Lin, et al. “CLIP is Also an Efficient Segmenter: A Text-Driven Approach for Weakly Supervised Semantic Segmentation”, CVPR, 2023
> > * [6] (OCR) Zesen Cheng, et al. “Out-of-Candidate Rectification for Weakly Supervised Semantic Segmentation”, CVPR, 2023.
> > * [7] (MMCST) Lian Xu, et al. “Learning Multi-Modal Class-Specific Tokens for Weakly Supervised Dense Object Localization”, CVPR, 2023.
> > * [8] (ACR) Hyeokjun Kweon, et al. “Weakly Supervised Semantic Segmentation via Adversarial Learning of Classifier and Reconstructor”, CVPR, 2023.
> > * [9] (BECO) Shenghai Rong, et al. “Boundary-Enhanced Co-Training for Weakly Supervised Semantic Segmentation”, CVPR, 2023.
> > * [10] (MARS) Sanghyun Jo, et al. “MARS: Model-agnostic Biased Object Removal without Additional Supervision for Weakly-Supervised Semantic Segmentation”, ICCV, 2023.
> > * [11] (USAGE) Zelin Peng, et al. “USAGE: A Unified Seed Area Generation Paradigm for Weakly Supervised Semantic Segmentation”, ICCV, 2023.
> > * [12] (FPR) Liyi Chen, et al. “FPR: False Positive Rectification for Weakly Supervised Semantic Segmentation”, ICCV, 2023.
> > * [13] (Mat-Label) Changwei Wang, et al. “Treating Pseudo-labels Generation as Image Matting for Weakly Supervised Semantic Segmentation”, ICCV, 2023.
> > * [14] (MCC) Fangwen Wu, et al. “Masked Collaborative Contrast for Weakly Supervised Semantic Segmentation”, WACV, 2024.
> >
> > We will add missing citations in the revised paper.

---

> > > ### Author Response · Authors · 2023-11-22
> > >
> > > We wish to emphasize that we have diligently addressed every comment. We would also be appreciative if you could consider re-evaluating the rating of this work. Please let us know if there are any concerns or questions, and we are eager to provide further clarification or discussion.

---

### Official Review · Reviewer_BYUr · 2023-11-02

**Soundness:** 4 excellent
**Presentation:** 3 good
**Contribution:** 4 excellent
**Rating:** 8
**Confidence:** 4

**Summary:**

A new Weakly-Supervised Semantic Segmentation (WSSS)  method, called SemPLeS, is proposed in this paper. In SemPLeS, Contrastive Prompt Learning and Class-associated Semantic Refinement are used to learn the prompts that adequately describe and suppress the image backgrounds associated with each target object category. The authors tested SemPLeS, and it outperformed the existing SOTA on popular benchmarks like PASCAL VOC and MS COCO.

**Strengths:**

Using contrastive learning and CLIP text/visual encoders is an interesting idea. Especially optimizing learnable negative prompts and applying it to contrastive learning with positive image regions is very interesting. The novelty of the proposed method seems to be high.

By the experiments, the effectiveness of the proposed method was clearly shown. The proposed method seems to outperform the current SOTA method, WeakTr.

**Weaknesses:**

According to Paper-with-code, SOTAs for val and test are 78.4 and 79.0 by WaekTr, respectively. Why the results of the baselines shown in Table 2 are less than that ? The authors should present results of the proposed method under the same settings as Paper-with-code SOTA.
https://paperswithcode.com/sota/weakly-supervised-semantic-segmentation-on-1
https://paperswithcode.com/sota/weakly-supervised-semantic-segmentation-on

**Questions:**

I'm wondering if the proposed method is also effective for panatomic segmentation tasks such as Pascal Panatomic and CityScapes.

---

> ### Author Response · Authors · 2023-11-16
> **Rebuttal**
>
> Thank you for all the constructive comments. Here are our replies:
>
> **1. SOTA comparison:**
>
> The better SOTA by WeakTr uses different settings, including a better ViT backbone pre-trained on ImageNet-21K [1] and a larger resolution (384) for fine-tuning. For the classification performance on the ImageNet-1K dataset, DeiT-S (our backbone) has 79.9 accuracy while ViT-S (SOTA WeakTr’s backbone) has 83.7 accuracy.
> We choose not to use the better WeakTr as the backbone because we would like to follow the standard protocol of the WSSS benchmark for fair comparison.
>
> **2. Panoptic segmentation:**
>
> To our best understanding, although there are some weakly-supervised panoptic segmentation papers like WSSPS [2] (box-supervised), PSPS [3] (point-supervised), and Panoptic-FCN [4] (point-supervised), there is no panoptic segmentation work with class supervision. It would be an interesting research direction to explore, and we see it as a potential future work.
>
> ---
> References:
> * [1] Andreas Steiner, et al. “How to train your ViT? Data, Augmentation, and Regularization in Vision Transformers”, TMLR, 2022.
> * [2] (WSSPS) Qizhu Li, et al. “Weakly- and Semi-Supervised Panoptic Segmentation”, ECCV, 2018.
> * [3] (PSPS) Junsong Fan, et al. “Pointly-Supervised Panoptic Segmentation”, ECCV, 2022.
> * [4] (Panoptic-FCN) Yanwei Li, et al. “Fully Convolutional Networks for Panoptic Segmentation with Point-based Supervision”, TPAMI, 2022.
>
> We will add missing citations in the revised paper.

---

> > ### Author Response · Authors · 2023-11-22
> >
> > We wish to emphasize that we have diligently addressed every comment. Please let us know if there are any concerns or questions, and we are eager to provide further clarification or discussion.

---

### Author Response · Authors · 2023-11-16
**Thank you for all the comments**

We sincerely thank all of the reviewers for their valuable comments and encouragement of the novel approaches (automatically learnable prompts for segmentation), thorough experiments that show method effectiveness, and clear explanations in the paper.
We provide answers to every comment, and we will further revise the paper accordingly.

---

### Author Response · Authors · 2023-11-21
**Revised Paper Uploaded**

We extend our sincere gratitude once again to all the reviewers for their invaluable feedback.  We uploaded the revised paper according to all recommended changes, where the revised parts are denoted by the use of the blue font for enhanced visibility. Furthermore, to enhance clarity and readability, we have made adjustments to the notation in Eq. (2) as follows:
* $L^b_{prompt}$ (old) → $L^I_{prompt}$ (revised): this loss aims to attract class-associated background prompts to predicted background *images*.
* $L^f_{prompt}$ (old) → $L^T_{prompt}$ (revised): the loss aims to repel class-associated background prompts from foreground class *labels*.

Some revised portions are located in the appendix due to the page limit.

We wish to emphasize that we have diligently addressed every comment provided by the reviewers. We eagerly await their feedback and are prepared to offer further clarification or address any additional questions or comments they may have.

---

### Meta-Review · Area_Chair_KkKH · 2023-12-05

**Metareview:**

This paper proposes a method for weakly-supervised semantic segmentation that uses CLIP embeddings to refine pseudo-masks by suppressing regions associated with learned background prompts and enhancing regions associated with prompts formed from foreground labels.

The paper received diverging ratings which remain after the discussion phase. All reviewers agreed that the proposed method is somewhat novel, and interesting. The main concerns were (i) insufficient comparisons to related methods; (ii) missing citations/discussions of related work; and (iii) the marginal performance gains, compared to SOTA, of the complex system being proposed. Reviewers are still concerned about lack of comparisons to related work that appeared on arxiv in April (Segment Anything, or SAM) and early May ([1], SAM-MFG [2]). The authors addressed the concerns about SAM by including a section in the revised Appendix discussing SAM. In this section they claim that because SAM was trained using annotated data, it's beyond the scope of this work. However, as the authors acknowledge in the discussion, SAM can and has been used to improve many image label-based WSSS methods. In particular, [1] achieves SOTA results in this way.

Because of these unresolved concerns, the majority recommendation from reviewers is to not accept. The AC agrees that concerns (i) and (iii) require more revision and suggest the authors  address them in a future version of this work.

[1] https://arxiv.org/abs/2305.01586
[2] https://arxiv.org/pdf/2305.11003

**Justification For Why Not Higher Score:**

See above

**Justification For Why Not Lower Score:**

n/a

---

### Decision · Program_Chairs · 2024-01-16

Reject